# WiPg: Contactless Action Recognition Using Ambient Wi-Fi Signals

**DOI:** 10.3390/s22010402

**Published:** 2022-01-05

**Authors:** Zhanjun Hao, Juan Niu, Xiaochao Dang, Zhiqiang Qiao

**Affiliations:** 1College of Computer Science and Engineering, Northwest Normal University, Lanzhou 730070, China; 201671020215@nwnu.edu.cn (J.N.); dangxc@nwnu.edu.cn (X.D.); 2019221817@nwnu.edu.cn (Z.Q.); 2Gansu Province Internet of Things Engineering Research Center, Lanzhou 730070, China

**Keywords:** device-free sensing, channel state information, human action standard recognition, personnel independence, generative adversarial network, principal component analysis

## Abstract

Motion recognition has a wide range of applications at present. Recently, motion recognition by analyzing the channel state information (CSI) in Wi-Fi packets has been favored by more and more scholars. Because CSI collected in the wireless signal environment of human activity usually carries a large amount of human-related information, the motion-recognition model trained for a specific person usually does not work well in predicting another person’s motion. To deal with the difference, we propose a personnel-independent action-recognition model called WiPg, which is built by convolutional neural network (CNN) and generative adversarial network (GAN). According to CSI data of 14 yoga movements of 10 experimenters with different body types, model training and testing were carried out, and the recognition results, independent of bod type, were obtained. The experimental results show that the average correct rate of WiPg can reach 92.7% for recognition of the 14 yoga poses, and WiPg realizes “cross-personnel” movement recognition with excellent recognition performance.

## 1. Introduction

With the widespread use of wireless networks and Wi-Fi devices, Wi-Fi has become a great green sensing platform, playing a decisive role in the field of perception. Previous studies on motion recognition can be roughly divided into three categories: wearable device-based, vision-based, and Wi-Fi-based [1]. Behavior-recognition technology based on wearable devices has many limitations, such as the additional increase in user burden, inconvenience in daily life, limited power of sensors, etc. Some articles require users to carry special devices [2,3,4], such as radio frequency identification (RFID) tags [5,6], gloves, or mobile sensors, which can be inconvenient and costly in many real-life scenarios. For example, MRLIGHT [5] installs RFID readers on moving objects (people) and fixedly deploys RFID tags in the monitoring area, effectively and economically realizing indoor real-time position tracking. However, a large number of fixed and passive RFID tags need to be deployed, and they are prone to environmental impact in MRLIGHT, which is accompanied by a lot of inconvenience. Although vision-based behavior-recognition technology can also achieve high recognition accuracy, it is affected by problems such as light, angle, privacy, and security. The authors used cameras to detect human falling behavior, designed an inference method based on two-dimensional CNN to obtain the motion information between two consecutive images, and tested the public data set, achieving high recognition accuracy [7].

Recently, an increasing number of people have been using inexpensive and simple-to-use commercial Wi-Fi devices for behavior recognition [8,9,10], including human-activity recognition, gait recognition, gesture recognition, and other related applications. Because the channel state information (CSI) of the physical layer can provide more fine-grained and accurate behavior data, it describes both the amplitude attenuation and the phase shift of wireless signals, according to which various behaviors, from vital signals and basic actions to complex activities, can be more effectively identified. Therefore, there are numerous research projects on behavior recognition based on CSI data.

For gait recognition, MAR [11] performs canonical polyadic (CP) decomposition on the signals reflected from different parts of the human body and recognizes gait according to the frequency and range of arms and legs (sometimes even the head). The uniqueness of CP decomposition is theoretically analyzed, and better accuracy of gait recognition is achieved. However, the limitation is that the anti-interference ability is weak. Multiple moving objects cause a multipath effect and other interference, which reduces the accuracy of MAR. Gate-ID [12] has proposed a method to infer people’s walking direction through gait characteristics and the direction of the antenna and has designed a deep learning model based on an attention and feature-weighting algorithm to identify individuals through CSI data containing gait characteristics. However, this article stipulated the walking path and does not attempt other random walk paths, which means the method in this paper is not universal.

There are also many studies on gesture recognition based on Wi-Fi. For example, SignFi [13] is able to identify 276 sign gestures with a high recognition rate by inputting CSI data containing gesture features in Wi-Fi packets into a nine-layer convolutional neural network for recognition. However, it can only recognize a single word but not a sentence in sign language. This paper collects and evaluates CSI data of 50 gestures from 5 users, extracts different levels of gesture features through a one-dimensional parallel long- and short-memory full convolutional network model, and is robust across different gestures and different people [14]. This point is similar to the point that this paper wants to study, which is of reference significance to us. However, this paper does not consider the robustness of different environments.

In a typical study of indoor behavior recognition based on Wi-Fi CSI, Hao et al. proposed a complex human-motion recognition scheme based on CSI [15], which is combined with a restricted Boltzmann machine (RBM) and SoftMax regression correction RBM classification. This method can classify the motion data sets effectively. However, this method is only aimed at static movements, while XingYiQuan motions are dynamic movements, and there is no research on dynamic movements. Another typical human-activity recognition method based on deep learning uses attention-based bidirectional long- and short-term memory to learn representative features of activities and actions from CSI sequences, which can obtain the best recognition performance of all activities [16]. However, this method does not take into account the effects of environmental changes. Although the above two papers on human-activity recognition achieved their goals and a high recognition rate, they did not pay attention to the influence of different people and did not conduct relevant studies on different people, so they are not universally applicable to people.

Based on the above research, we consider that whether it is gesture recognition, gait recognition, or indoor activity recognition, in addition to trying various devices to improve the accuracy of identification, it is also necessary to consider the universal availability of identification methods. For example, it is worth considering whether the model trained according to the CSI data of specific experimenters can recognize the actions of other experimenters. Since the focus of our study is on human-independent motion recognition methods, we also studied the latest papers in this respect. For the sake of solving the problem of activity identification independent of people and environment, there are different methods for research. Jiang et al. [17] raised a series of questions. For example, the reflection of the wireless signal in space is affected by various environmental factors (such as glass, air, walls, etc.), resulting in a change in the reflected signal. There are also significant differences in the reflected signals received when detecting the activity of different user objects, such as adults and children with wide differences in height and weight. These problems result in activity-recognition devices that do not work well in the new environment. An activity-recognition method independent of humans and the environment is realized through Doppler translation and neural network model construction in this study. However, the recognition rate of other data sets is not very high. Ma et al. [18] built an activity-recognition model by combining the theories of the state machine and reinforcement learning. They trained people-independent and location-independent recognition models by learning location and people’s independent features from different angles of CSI and evaluated the models through multiple experiments and public data sets, with quite good results can be obtained by their work. However, it does not necessarily work in multiplayer scenarios. Widar3.0 [19] can extract domain-independent data features that only reflect the gesture itself from the original domain-dependent CSI signal by establishing a quantitative relationship model between complex gestures and CSI dynamics. Cross-domain gesture recognition is realized by the generalization technique, and the accuracy of gesture recognition based on CSI was improved to 92.4% in the cross-environment situation. The above three articles have reference significance for this paper.

As we all know, with improvement in living standards, people’s work pressure is heavier and heavier. Because of work, many people suffer from a variety of “occupational diseases” at present, such as cervical spondylosis, periarthritis of the shoulder, etc. However, studies have shown that many of these “occupational diseases” can be alleviated or even cured by practicing related yoga poses, such as the half moon, which can relieve sciatica caused by sitting too long. At the same time, yoga can play a role in weight loss, shaping, regulating emotions, relieving pressure, and preventing cardiovascular and cerebrovascular diseases, as well as other aspects. However, to achieve the purpose of the exercise is the premise of yoga movements. In real life, in addition to yoga classes under the guidance of teachers, students learn by watching videos on their own, and in many cases, training results are counterproductive. Based on this background, we take CSI data of yoga actions as the data sample of action recognition in this study, which has highly practical significance. The yoga poses we adopt in this study are shown in Figure 1. The difference between the body types of different people will affect the performance of the recognition method, so the recognition method in this paper should be suitable for different people.

This study solves the above problems by building a deep learning model that can effectively identify yoga movements. We obtained an action-recognition method independent of human, and a feedback update module is introduced into the model. The output results that are not within the threshold, which is set in advance and obtained experimentally, are fed back to the model for retraining. If the final output is obtained after repeated training, the action is judged to be the standard action. WiPg will provide an effective scientific guide for the user’s family exercise. Similarly, Qi et al. [20] designed a multilayer recursive neural network to identify teleoperated robot gesture data obtained by multiple sensors, thus guiding human-machine collaboration tasks. The proposed method has a high recognition rate, but the data-acquisition system has limitations. Su et al. [21] modeled the dynamic motion primitive (DMP) during the operation of experienced surgeons by teaching by demonstration (TbD) technology to help surgical robots learn surgical skills and complete surgical tasks. The operational autonomy of the surgical robot is increased, and the work efficiency of the surgeon is improved, but this article only considers surgical skills, not operational strength. Qi et al. [22] combined deep learning and threshold methods to monitor breathing and motion recognition, and compared with previous machine learning algorithms, they achieved a higher recognition rate and more robustness, although the ability to analyze breathing patterns is limited. These articles are based on the learning of real-life activities, and the learning results have a guiding role in related applications, the same intention as our article. In the last paper, the combination of deep learning and threshold value was used to solve the identification problem, which indicates that the idea proposed in WiPg is feasible and has certain reference value for us to use the deep learning model for identification.

Furthermore, “Wi” refers to Wi-Fi based, “P” refers to personnel-independent, and “g” refers to yoga in the word “WiPg”. Three contributions of WiPg are as follows:Aiming at the problem that the diversity between the body types of various people affects the performance of the recognition method, the recognition method in WiPg is suitable for diverse people. Through the combination of GAN and CNN, high-level features independent of body size are extracted, and action recognition is carried out.WiPg has a very good recognition accuracy. In order to achieve this goal, first of all, we need to collect a large number of real experimental data and create similar graphs of problems by using different numbers of samples. More samples provide more opportunities for learning algorithms to understand the underlying mapping of input to output, leading to better performance models. Secondly, in the process of model training, parameters are adjusted repeatedly until the best recognition result is obtained.On account of the longer training time, people who practice yoga exercises cannot maintain the standard movements of the initial state from the beginning to the end of the whole training process. This can be judged by setting the threshold value in WiPg. If multiple feedback updates fail to output the result, it indicates that the action is not standard; and if it can output a classification result, it indicates that the action is standard.

## 2. Related Theory

### 2.1. Theory of CSI

CSI signals are acquired from ubiquitous Wi-Fi devices and reflect the attenuation of Wi-Fi signals in transit. The CSI signal of yoga action received by the receiver can be expressed as:(1)$$H\left(f_{k}\right)=\left\|H\left(f_{k}\right)\right\|e^{j∠H\left(f_{k}\right)},k\in \left[1,K\right]$$
where $H\left(f_{k}\right)$ is CSI with $f_{k}$ as the center frequency representing the kth subcarrier of the CSI signal of yoga action;$\left\|H\left(f_{k}\right)\right\|$ is amplitude; and $∠H\left(f_{k}\right)$ is phase. If $H_{k}=\left\|H\left(f_{k}\right)\right\|$, the yoga CSI sequence of N pairs of antennas is expressed as Equation (2):(2)$$H^{n}=\left\lfloor H_{1}^{n},H_{2}^{n},\ldots ,H_{k}^{n}\right\rfloor ,n\in \left[1,N\right]$$
where $H_{k}^{n}$ stands for the *k*th subcarrier of CSI sequence between the nth pair of antennas. For the sake of identifying yoga movements in the detection area, we need to regularly collect continuous CSI data and take M sets of CSI sequences collected within a period of time as a group. The CSI sequence can be expressed as Equation (3):(3)$$\left[\begin{array}{cccc} {H^{\prime }}_{11} & {H^{\prime }}_{12} & \cdots & {H^{\prime }}_{1k}\\ {H^{\prime }}_{21} & {H^{\prime }}_{22} & \cdots & {H^{\prime }}_{2k}\\ \vdots & \vdots & \ddots & \vdots \\ {H^{\prime }}_{m1} & {H^{\prime }}_{m2} & \cdots & {H^{\prime }}_{mk} \end{array}\right],m\left(1,M\right),k\left(1,K\right)$$
where ${H^{\prime }}_{mk}$ stands for the set of the *k*th subcarrier at the *m*th time. CSI signals are affected by human activities in Wi-Fi-covered areas, as shown in Figure 2, because the human body may block the transmission path of Wi-Fi wireless signals and thus cause signal power attenuation. Otherwise, the human body can introduce more signal reflections and change the number of propagation paths. Therefore, the variance of CSI can reflect human movement in the wireless environment. Therefore, CSI signals are used to reflect the differences of different yoga movements and predict movements in WiPg.

### 2.2. Methods of Data Preprocessing

#### 2.2.1. An Overview of the Butterworth Filter

We selected the Butterworth filter to preprocess the collected CSI data related to yoga in this study. The filter is introduced in detail below. The amplitude square function of the Butterworth filter amplitude is shown in Equation (4).
(4)$${\left|H_{a}\left(j\Omega \right)\right|}^{2}=\frac{1}{1+{\left(\frac{\Omega }{\Omega_{c}}\right)}^{2N}}$$
where *N* is the order of the filter, Ω is the cut-off frequency, and $\Omega_{c}$ is the cut-off frequency at −3dB [23]. It can be seen from the above formula that the n-order filter has 2*N* poles, and these poles are distributed in the Butterworth circle. When its poles are located in the left half plane of the S plane—that is, *N* poles in the left half plane are taken as the poles of the filter—the filter is stable. Before calculating the stable pole, we need to calculate the stopband attenuation through Equation (5) and calculate the order through Equation (6) at first. Then, we should obtain the stable pole through Equation (7).
(5)$$A_{s}=-20\log_{10}\left|H_{a}\left(j\Omega_{s}\right)\right|$$
(6)$$N=\frac{1}{2}\frac{\log_{10}\left(\frac{10^{0.1A_{s}}-1}{10^{0.1A_{s}}-1}\right)}{\log_{10}\left(\frac{\Omega_{s}}{\Omega_{p}}\right)}$$
(7)$$P_{k}=\left\{\begin{array}{l} \Omega_{c}e^{j\frac{2k+1}{2N}\pi },N\mathrm{is}\mathrm{even},k=0,1,2,\ldots ,2N-1\\ \Omega_{c}e^{j\frac{2k}{2N}\pi },N\mathrm{is}\mathrm{odd},k=0,1,2,\ldots ,2N-1 \end{array}\right.$$
where $A_{s}$ is the stopband attenuation and *P_k_* is the stable pole. Then, the transfer function is calculated, as shown in Equation (8).
(8)$$H_{a}\left(s\right)=\prod_{Re\left[P_{k}\right]<0}\frac{\Omega_{c}}{s-P_{k}}$$
where $H_{a}\left(s\right)$ is the coefficient of the analog filter. Then, the variable, *s*, of $H_{a}\left(s\right)$ in Equation (8) is replaced by Equation (9) through a bilinear transformation from the analog domain to the digital domain, and the coefficient, $H\left(z\right)$, is obtained, as shown in Equation (9). Finally, the filtering result can be calculated through the differential equation.
(9)$$s=\frac{2}{T}\frac{1-z^{-1}}{1+z^{-1}}$$

#### 2.2.2. An Overview of the PCA

As a basic dimension-reduction technique, principal component analysis (PCA) removes the redundancy of original data and maintains the inherent structure of original data after dimensionality reduction [24]. In order to remove redundancy, the original CSI data should be projected to the direction with the maximum variance because the greater the variance, the greater the difference between the data, so that the redundancy between the data will be smaller. The algorithm principle is described, as follows, with mathematical knowledge:

Assuming that there are n m-dimensional sample data sets,$D=\left\{x_{1},x_{2},\ldots ,x_{n}\right\}$, $x_{i}\left(1\leq i\leq n\right)\in R^{m}$, and assuming that the characteristics of each dimension have been normalized, if the dimension of *x_i_* is reduced from *m* to *l* and the data after dimension reduction are represented by Y, a mapping matrix, W, is required so that W satisfies the following formula.
(10)Y=XW

In this formula, the reduced dimension is denoted by Y. Since X is an *n*m* matrix and y is an *n*l* matrix, W should be an *m*l* matrix. The variance of Y is expressed as follows:(11)$$\mathrm{var}\left(Y\right)=\frac{1}{n-1}trace\left(Y^{T}Y\right)=trace\left(W^{T}\frac{1}{n-1}X^{T}\mathrm{XW}\right)$$
(12)$$\sum =\frac{1}{n-1}X^{T}X$$
where ∑ is the covariance matrix of the n m-dimensional sample, X, before dimensionality reduction. In order to maximize the variance of Y, $trace\left(W^{T}\sum W\right)$ should be maximized. In order to reduce the correlation—that is, to remove redundant data—W needs to satisfy $W_{i}{}^{T}W_{i}=1$, $i\in \left(1,2,\ldots ,l\right)$, and then it can be converted into Equation (13) by the Lagrange multiplier method.
(13)$$L\left(W,\lambda \right)=trace\left(W^{T}\sum W\right)-\sum_{i=1}^{l}\lambda_{i}\left(W_{i}^{T}W_{i}-1\right)$$
where $\lambda_{i}\left(1\leq i\leq l\right)$ is the Lagrange multiplier and W*_i_* is column *i* of the matrix, W. Take the partial derivative of *W_i_* and set the derivative to 0, and you get the following formula.
(14)$$\sum W_{i}=\lambda_{i}W_{i}$$
(15)$$trace\left(W^{T}\sum W\right)=\sum_{i=1}^{l}\lambda_{i}$$

As can be seen from the above equation, the optimized variance is equal to the sum of the characteristic roots of the covariance matrix of the original sample data. Therefore, the mapping matrix, *W*, can be composed of the eigenvectors corresponding to the first *l* largest characteristic root.

According to the principles of the two data pretreatment methods described in Section 2.2, the more important action feature data in this study and CSI data can be obtained, and the integrity of the CSI signal can be retained.

### 2.3. Overview of the GAN

A generative adversarial network (GAN) is composed of two important parts: discriminator network and generator network [25]. The goal of the discriminator network is to distinguish whether *X* comes from a real distribution or a generated model, which is actually a binary classifier. Conversely, the generator network needs to let the discrimination network distinguish the samples generated by itself into real samples. The basic structure of GAN is shown in Figure 3.

*X* represents the real data, *Z* represents the noise of the generator network, $G\left(z\right)$ means unreal data generated by the generator network, and $D\left(x\right)$ represents the probability that *x* belongs to the real sample distribution, where $D\in \left[0,1\right]$. The optimization principle of GAN is simply that the generator network, *G*, generates $G\left(z\right)$ through continuous training and learning and makes the discriminator network, *D*, unable to distinguish the difference between $G\left(z\right)$ and *x*. *D* is to improve their discriminant ability through continuous training and learning, that is, to recognize that *x* and $G\left(z\right)$ are different.

The optimization principle of GAN can be described by a mathematical formula, as shown in Equation (16):(16)$$\underset{G}{min}\underset{D}{max}V\left(D,G\right)=E_{x\sim P_{data}\left(x\right)}\left[logD\left(x\right)\right]+E_{z\sim P_{z}\left(z\right)}\left[log\left(1-D\left(G\left(Z\right)\right)\right)\right]$$

Firstly, *D* is optimized, and then *G* is optimized. Optimization is about fixing one side and training the other. The purpose of *D* is to correctly distinguish *x* and $G\left(z\right)$. When optimizing the discriminator network, *D*, *G* should be given in advance, and efforts should be made to increase $D\left(x\right)$ and decrease $D\left(G\left(z\right)\right)$. Therefore, the discriminator network is optimized is to get $\underset{D}{\max}V\left(D,G\right)$.

Similarly, when optimizing the generator network *G*, *D* needs to be given in advance, and we just need to increase $D\left(G\left(z\right)\right)$. The generation network is optimized to get $\underset{G}{min}V\left(D,G\right)$.

According to the above antagonism principle of GAN, in order to remove the influence of the body size of personnel, the collected data can be trained through the mutual antagonism mode of the two networks in WiPg. The two networks are updated and iterated many times to maximize the interference ability of *G* to *D* and to minimize the discrimination ability of *D* to personnel tags. In the end, *G* makes it impossible for *D* to tell who the action is generated by, while the discriminator network, *D*, tries to tell n the action is generated by, so as to solve the personnel independence of action recognition.

## 3. Overview of WiPg

The overview of WiPg is shown in Figure 4. We selected recently popular yoga movements as the research object in this study, and the steps of the identification method are as follows:We collected the CSI sign of 14 standard yoga poses from 10 experimenters in three real experimental environments. We eliminated the noise of the collected data by Butterworth filter and PCA, and the main characteristics of the yoga CSI data were retained. See Section 3.1 for details;According to fast Fourier transform, energy changes to determine which period of data is yoga action data. See Section 3.2 for details;CNN is integrated into GAN to build an action-recognition model through which to learn a piece of yoga CSI data that is personnel-independent, followed by action recognition. See Section 3.3 for details;The last step is to estimate whether the yoga action is standard or not. The criterion is the threshold value obtained through a large number of training tests. If the output is within the threshold range, the action is standard, and the recognition result will be output. On the contrary, if it is not in the range, feedback is updated, and the action is not standard if it is in a loop in which the result cannot be output all the time. See Section 3.3 for details.

### 3.1. Data Preprocessing

On account of the impact of Wi-Fi equipment and the interference of environmental factors, there is a lot of interference and noise in the collected CSI data. To obtain high-quality data, we need to remove noise from the original CSI data. For this reason, to obtain more accurate motion features in CSI data, the Butterworth filter is firstly carried out on the original data to remove high-frequency noise. Secondly, because of the large amount of CSI data that belongs to the high-dimensional data, the PCA algorithm is applied to reduce the quantity of the original CSI data and retain the most crucial feature data. According to the principles of the two pretreatment techniques described in Section 2.2, the original and the preprocessed waveform are shown in Figure 5. In Figure 5a, there are three antennae from top to bottom, and each antenna contains 1900 CSI data. *n* curves with different colors represent the subcarrier amplitude variation curve on the *n*th packet. Figure 5a contains three antennas, and the total number of *n* is 3*1900.

It can be seen in Figure 5a that a large amount of noise is embraced in the original signal. According to the principles of the two methods described in Section 2.2, a Butterworth filter is first applied to remove high-frequency noise. We adopted PCA to lessen the dimensionality of superfluous data characteristics. At the same time, it retains the most substantial bits of the data characteristics and maintains the data integrity to the greatest degree. Figure 5b shows that the dimension of the CSI data, treatment by PCA, is reduced, and the prime features of yoga-movement CSI data are retained, which is fundamental to the subsequent feature extraction.

### 3.2. Motion Detection

Wi-Fi devices always exist in the real scene, so once people enter this environment, CSI data related to the human body are generated. Therefore, CSI data collected in the real scene include many other human-behavior data unrelated to yoga actions. To solve this problem, we need to judge these CSI data and find out which are the CSI data related to yoga movements required in WiPg. Different human behaviors cause different energy changes in the CSI frequency domain. Fast Fourier transform (FFT) can transform the CSI signal after eliminating the noise from the time domain to the frequency domain [26]. Then, by analyzing the CSI signal of yoga movements in the frequency domain, an energy threshold can be obtained to judge the occurrence and end of movements. Figure 6 is the energy change in the frequency domain caused by POSE TWO in Figure 1.

The energy distribution in the frequency domain after FFT transformation is shown in Figure 6. It can be seen that after normalized FFT conversion, there are two large peak signals, namely the 6 Hz signal and the 10 Hz signal. This segment of signal is the dynamic frequency variation range caused by POSE TWO. The frequency variation caused by DC signal is not recorded in the frequency variation range caused by the action. The FFT peak value of the blue circle in Figure 6 is lower than 1, which is considered to be the controllable signal caused by POSE TWO and is not recorded as the frequency range caused by the action. Therefore, we can use the signals between the two larger peaks to ensure that CSI can be collected effectively when the action occurs. This will be of great help to the subsequent feature extraction.

### 3.3. Action Recognition of Wi-Piga

A network GAN, including CNN, which is regarded as a feature-extraction algorithm, takes the central stage in WiPg. The personnel feature-discrimination module is defined to obtain a personnel-independent identification result, which is equal to the discriminator network, *D*. Figure 7 shows a concrete structure.

The training process of GAN in WiPg is as follows:To generate parameters of network *G* and discriminator network, *D*, by initialization;To extract *n* real data samples with personnel labels from the training set, *G* generates the personnel labels of the *n* data sample without personnel labels and forms new data;To fix *G* and then to train *D*, we make *D* distinguish the true and false data samples as much as possible; in other words, to mark the person who generated each CSI data and to obtain a personnel-label distribution, R, through full connection;To update *D* for *k* times, *G* is updated once. *D* maximizes the prediction performance of personnel labels, and *G* minimizes the prediction accuracy of discriminator model, *D*. After multiple update iterations, the people-related features are ideally eliminated, and the performance of person tag prediction is reduced.

The concrete implementation of the generation network and the discriminator network will be introduced in the following sections.

#### 3.3.1. Generation Network, *G*

The Generation network, *G*, is divided into two parts. The first is the feature-extraction module, where we mainly adopt a convolutional neural network for yoga-movement data of CSI signal feature extraction. Due to CNN having been widely used for human-activity recognition feature extraction [27], this article also directly uses CNN to extract the features. Three layers of CNN were used in this article. Each layer of CNN contains a 2D kernel as a filter and then normalizes the mean value and variance of the data in each layer.

Finally, we add a ReLU and a maximum pooling layer to make the features between different dimensions have some numerical comparison. The result, *S*, after CNN convolution is shown in Equation (17).
(17)$$S=\mathrm{CNN}\left(X;\theta \right)$$
where *S* is the feature set, *X* is the input, and *θ* is the parameter set of CNN. The input data of the feature-extraction module is *X*, and all the data of *X* are marked. Each piece of data, *X_i_*, has a corresponding feature-label vector, *Id_i_*, related to personnel, where $Id_{i}\in D$, *D* represents all the *Id_i_* sets, and each piece of data, *X_i_*, corresponds to a real vector,$y_{i}$.

The second part is called the action-recognition module. Firstly, as shown in Equation (18), the full connection layer and an activation function are used to obtain a feature representation after learning. The softplus function is an activation function that introduces nonlinearity. $F_{i}$ is then mapped into a new space, $H_{i}$, where $H_{i}\in R^{C}$, *C* is the amount of action data. The softmax layer is used to obtain the probability vector of the activity, as shown in Equation (19).
(18)$$F_{i}=Softplus\left(W_{S}S_{i}+b_{S}\right)$$
(19)$${\overset{⌢}{y}}_{i}=Softmax\left(H_{i}\right)andH_{i}=W_{f}F_{i}+b_{f}$$
where $W_{s}$ is the weight matrix of the feature set, *S*; $b_{s}$ is the deviation of the feature set, S; $W_{f}$ is the learned feature representing the weight matrix of *F*; and *b_f_* is the learned feature representing the deviation of *F*. The cross-entropy function is used to calculate the loss, *L_a_*, between the predicted result, $\overset{⌢}{y}$, and the actual result,y, as shown in Equation (20).
(20)$$L_{a}=-\frac{1}{\left|X\right|}\sum_{i=1}^{\left|X\right|}\sum_{C=1}^{C}y_{iC}\log\left({\overset{⌢}{y}}_{iC}\right)$$

#### 3.3.2. Discriminator Network, *D*

The discriminator network, *D*, is constituted by the personnel-feature-discrimination module. The goal of *D* is to identify and record the label of the person who produces the yoga action, so as to force the feature-extraction module to produce yoga action features unrelated to the personnel. We first connect the output, S, of feature-extraction module with prediction $\overset{⌢}{y}$ to get U; that is, $U=S⊕\overset{⌢}{y}$, where ⊕ is the joint operation, and S contains both human-related and human-independent features. To identify the common properties between different people, we connect the above two parts as the input of this module and then obtain a label distribution related to people by using two fully connected layers with an activation function, as shown in Equations (21) and (22).
(21)$$T_{i}=Softplus\left(W_{u}U_{i}+b_{u}\right)$$
(22)$$R_{i}=Softmax\left(W_{t}T_{i}+b_{t}\right)$$
where $W_{u}$ is the weight matrix of U; $b_{u}$ is the deviation of U; $T_{i}$ represents the potential space; $W_{t}$ is the weight matrix of T; and $b_{t}$ is the deviation of T, in order to enable the module to identify the personnel-related characteristics of the input data. In WiPg, the cross-entropy function is used to calculate the loss, $L_{b}$, between the predicted personnel-related label distribution, R, and the actual personnel-related label vector, Id, as shown in Equation (23).
(23)$$L_{b}=-\frac{1}{\left|X\right|}\sum_{i=1}^{\left|X\right|}\sum_{j=1}^{\left|D\right|}Id_{ij}log\left(R_{ij}\right)$$
where $\left|D\right|$ is the number of features associated with a human; $Id_{i}$ is the actual personnel-tag vector. The goal of *D* is to minimize the loss function,$L_{b}$, so as to maximize the performance of human-label prediction, which is a contradiction for the final goal of WiPg, which is solved by Equation (24).
(24)$$L=L_{a}-\beta L_{b}$$
where β is the weight value. Finally, through Equation (24), we can observe that *G* tries its best to deceive *D* by maximizing $L_{b}$ while improving the performance of the action-recognition module by minimizing $L_{a}$ so that we can learn all of the characteristics that are common to people, and recognition results can be obtained. By inputting the standard action data into the above two networks for training, the threshold, *TH*, of the loss value, *L*, is obtained. Then, we input CSI data of an action into the WiPg network for testing. First, a prediction result, $\overset{⌢}{y}$, is obtained. Then, the loss value, *L*, is obtained. By comparing *L* with *TH*, if *L* is in *TH* interval, it means that it is standard, and the prediction result, $\overset{⌢}{y}$, is output. On the contrary, if *L* is in *TH* interval, the network will be updated by feedback. If the predicted action type cannot be output all the time, it means that the action is not standard.

## 4. Experiment and Analysis

### 4.1. Experimental Scene

In the experiment of WiPg, we use two laptops equipped with an Intel Wi-Fi link (IWL) 5300 network card as the receiver and transmitter of CSI packets, separately. We use the Linux CSI tool to extract CSI data from the network card to obtain the original CSI data containing yoga actions. The height of the equipment and the range of different yoga movements considered comfortable by different experimenters was counted and calculated. The average height of the equipment is 1 m, and the average distance of the equipment is 1.7 m.

However, in order to verify the performance of WiPg in different environments, CSI data in three experimental environments were collected for comparison during the experiment. Figure 8a is the plane structure of the yoga classroom. In this scene, the site is relatively empty, with fewer indoor items and less human interference, and the size of the yoga classroom is 10 m × 7 m. Figure 8b shows the plane of the laboratory. The size of the lab, which has more materials than the yoga classroom, is 7 m × 6 m. Figure 8c shows the plane of the student dormitory with the largest interference from the environmental multipath effect compared with the first two, with a size of 6 m × 4 m.

During CSI data collection, subjects posed in standard yoga poses between two Wi-Fi devices under the supervision of a yoga teacher. In WiPg, five males and five females (aged from 12 to 27 years old) were invited to collect the CSI data of yoga movements. As shown in Figure 1, each experimenter was asked to do 14 sets of yoga poses. We set different packet-sending rates and packet numbers in advance. The experimenter kept still for a period of time after doing yoga movements, and each movement needed to be repeated 20 times. Table 1 shows the body characteristics of the experimental participants, including gender, height, weight, and BMI (body mass index).

It is worth noting that the static environment may change over several days, and there may be more or fewer differences between CSI data collected by different movements of the same experimenter. Therefore, we collected CSI samples of yoga movements within nearly a month, which can commendably refrain from overfitting and enhance recognition ability. In total, more than 2000 CSI data streams related to yoga poses were collected.

### 4.2. Performance Evaluation

As is known to all, the common evaluation indexes of deep learning algorithms include confusion matrix, accuracy, ROC (receiver operating characteristic), AUC (area under curve), precision and recall, etc. The confusion matrix, which is also known as the error matrix, has each column representing the predicted value and each row representing the actual category [28]. F1 score is the harmonic mean of accuracy rate and recall rate. They are defined as follows:(25)$$\mathrm{Pr}ecision=\frac{TP}{TP+FP}$$
(26)$$\mathrm{Recall}=\frac{TP}{TP+FN}$$
(27)F1 Score=2*Precision*RecallPrecision+Recall
(28)$$\mathrm{Accuracy}=\frac{TP+TN}{TP+TN+FP+FN}$$
where *FN* is considered a negative sample, although it is, in fact, positive. *FP* is judged to be a positive sample, although it is, in fact, negative. In the same way, *TN* is judged to be a negative sample, which is correct. *TP* is judged to be a positive sample, which is correct.

### 4.3. Experimental Verification

#### 4.3.1. Influence of Different People’s Bodies on the Recognition Rate

In this study, we mainly discuss the influence of body size on CSI data and how to deal with this influence. Therefore, CSI data collected for people with different body types are tested in this section. Figure 9 shows the original CSI waveform was obtained by three experimenters with differing body types. There are three antennae from top to bottom, and each antenna contains 1900 CSI data. *n* curves with different colors represent the subcarrier amplitude variation curve on the *n*th packet. Figure 9 contains three antennae, and the total number of *n* is 3*1900. Figure 10 is the CSI waveform after pretreatment of the original waveform in Figure 9. CSI data collected by six experimenters with different body types were specially selected, and the experimental results obtained by training were input into the model in the three experimental environments, as respectively shown in Figure 11.

It can be seen from the original CSI waveform in Figure 9 and the preprocessed CSI waveform in Figure 10 that the signal changes are different when people of different body types do the same movement in the yoga classroom, which indicates that the different body type of experimenters has a an impact on CSI data and therefore on the identification results of subsequent experiments. This is consistent with the actual situation.

However, as shown in Figure 11, when in the same environment, the experimental results obtained by different experimenters doing the same action are similar. At the same time, in combination with the contents of the above three figures, it can be shown that the identification method of WiPg can ensure that the body shape of personnel does not affect the experiment; that is, it proves that WiPg is personnel-independent.

In the experiment in Figure 11, the first four yoga poses in Figure 1 were selected. It can be seen that the experimental results of POSE THREE are generally good, which is reasonable. Due to the large amplitude of POSE THREE, the characteristics of POSE THREE in the obtained CSI signal are more obvious, so the recognition results are relatively high.

#### 4.3.2. Impact of Different Number of Packets, Number of Subcarriers, and Experiment Scenes on the Recognition Rate

Different numbers of CSI packets transmitted and selected subcarriers will also have a certain influence on the recognition results because the small number of subcarriers is usually not a good representation of all the information contained in the CSI data. Therefore, we set up the above two comparative experiments in three different experimental scenarios in particular, and the recognition results are shown in Figure 12 and Figure 13.

It can be clearly seen from Figure 12 and Figure 13 that in three experiments, when the number of subcarriers is selected as three and the total number of transmitted packets is 5000, the model shows the best recognition effect. Therefore, the selection of appropriate experimental setting parameters (such as the above two) is very helpful to the model recognition rate. At the same time, it can also be seen that the recognition results in the yoga classroom are better than those in the other two environments, which is reasonable because the empty yoga classroom has the weakest multipath effect and the least interference of various environmental factors. Similarly, CSI signals collected in student dormitories are most affected by the multipath effect, so the recognition rate is the lowest compared with the other two.

#### 4.3.3. Impact of the Number of Epochs and Batch Size

When a complete data set passes through the neural network once and returns once, this process is called an epoch. In other words, all training samples have a forward propagation and a backpropagation in the neural network. In the process of neural network training, it is not enough to iteratively train all the data once [29]. When all the data of an epoch is trained once, each data generate loss, which updates the weight of the network. The updated network has a better fit for the data, so the result curve gradually changes from under-fitting to fitting. However, when the number of epochs is too large, the resulting curve is too fitting. In layman’s terms, the network now fully remembers the training data and is no longer portable.

Therefore, in this study, different numbers of epochs were set for model-training experiments so as to select the best number of epochs for neural network training in WiPg, as shown in Figure 14.

However, when an epoch (that is, all training samples) is massive for the computer, it needs to be divided into multiple small batches for training [30]. Therefore, batch size is also an important parameter in machine learning. The appropriate batch-size range is mainly related to convergence speed and stochastic gradient noise. If the batch size is too small and there are many categories, the loss function may oscillate, which is not conducive to convergence, especially when your network is complex. If the batch size is too large, the gradient direction of different batches does not change, and it is easy to fall into a local minimum. The optimization and generalization of deep learning has problems. Therefore, an appropriate batch size is very important for the performance of the model. In order to discuss the impact of batch size on the accuracy of the method, we set different batch sizes and, in turn, set batch size to improve the model performance. The result is shown in Figure 15.

It can be seen from Figure 14 and Figure 15 that the appropriate number of epochs and correct batch sizes are of great help to the performance of the model during model training. Because the diversity of data sets determines the number of epochs, the stronger the diversity, the larger the epoch should be. Therefore, for the 14 categories of collected data sets in WiPg, the larger the value of epoch, to a certain extent, the better the performance of the model. However, an appropriate batch size needs to be selected so that the model is not under-fitted or over-fitted. In general, in this study, when the number of epochs is 100 and the batch size is 256, the performance of the model reaches the optimum.

#### 4.3.4. Comprehensive Evaluation

The experimental parameters with the highest recognition rate can be obtained by comparing the above experiments. The specific identification method is described in Section 3.3. The comprehensive identification results of WiPg are shown in Figure 16.

Each column of the confusion matrix represents the prediction category, and the total number of each column represents the number of data predicted to be in this category. As shown in the above confusion matrix, it can be seen that the comprehensive recognition result reaches more than 90%, which means that WiPg can effectively recognize and distinguish the 14 yoga poses.

(1)Comparative experiments of different classification methods

Moreover, in order to verify the overall performance of the WiPg method, we conducted comparative experiments with support vector machine (SVM) and random forest (RF), while the evaluation index was accuracy, F1 score, and receiver operating characteristic (ROC) curve. The experimental results of the three methods are shown in Table 2. In order to more intuitively show the superiority of the method of WiPg, an ROC curve was set, as shown in Figure 17.

As can be seen from above, the accuracy of the three methods can reach more than 83%, but compared with the other two methods, the accuracy of the WiPg is higher and relatively stable, and the overall recognition performance is better.

For the CSI data collected in the yoga classroom, the accuracy of all three methods reaches more than 89%. However, the WiPg method has the highest rate of accuracy, and through the value of the F1 index, AUC area, and ROC curve, we can see that the performance of the algorithm in this paper is higher and the robustness is better. Therefore, the method presented in WiPg is more suitable for the recognition of yoga movements than the other two methods.

(2)Comparative experiments of different motion-recognition methods

At the same time, in order to further analyze the robustness of WiPg, we also compared WiPg with two other action-recognition methods (CSI-HC [15] and ABLSTM [16]). Similarly, we selected the three experimental environments mentioned above and 14 yoga action data sets collected by us for this comparative experiment. The evaluation results are shown in Figure 18 and Figure 19.

Figure 18 verifies the applicability of the three methods in different environments. It can be seen that the identification accuracy of WiPg is the highest. Figure 19 shows the evaluation of the above three methods in precision, recall, and F1 score, and the results show that the effectiveness of the proposed method is superior to that of the other two methods. Through a series of comparative studies above, we effectively proved the stability and higher classification accuracy of the WiPg deep classification model.

## 5. Conclusions

In allusion to the matter of action recognition based on Wi-Fi, we have proposed a personnel-independent deep learning method. A mass of CSI data related to yoga poses was collected in three real environments, and the original CSI data were preprocessed by Butterworth filter and PCA. Next, CNN and GAN were combined to obtain an action-recognition model independent of human body size. The experiment proved that the recognition method proposed in WiPg is personnel-independent, and WiPg has a good performance for yoga action recognition. At the same time, WiPg uses thresholds from a large number of experiments to determine whether the yoga poses are standard or not.

This study will be beneficial to family fitness exercise guidance, and the following point can be further studied, as follows: on the basis of judging whether a yoga movement is standard, we found a method to identify the standard degree of yoga movement more accurately, which can be used in home yoga monitoring systems in the real life.

However, our paper has the following limitations. At present, the model in WiPg only studies yoga movements and does not consider the migration effect of the model on other CSI data sets (such as gestures and falls), as well as the generalization ability of other data sets. Therefore, we believe that further research can be carried out on the following two points in the future: (1) On the basis of judging whether yoga poses are standardized, we can identify the degree of standardization of yoga poses in a more fine-grained way from the perspective of the differences between non-standard data and standard data; (2) The generalization ability of the model can be improved by training the network through data enhancement or using open-source data sets.

## Figures and Tables

**Figure 1 sensors-22-00402-f001:**
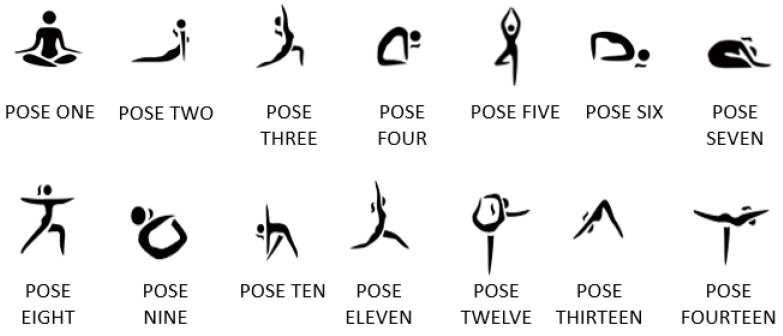
Yoga movements in WiPg.

**Figure 2 sensors-22-00402-f002:**
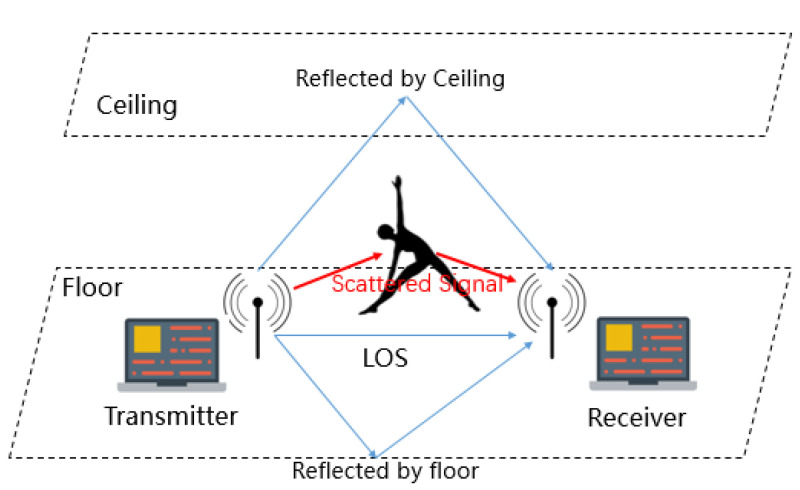
Wi-Fi signal propagation during human movement.

**Figure 3 sensors-22-00402-f003:**
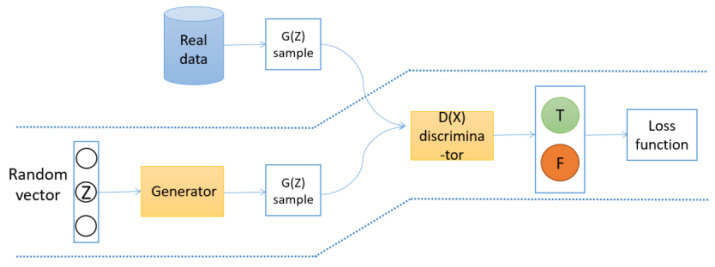
Overview of the GAN.

**Figure 4 sensors-22-00402-f004:**
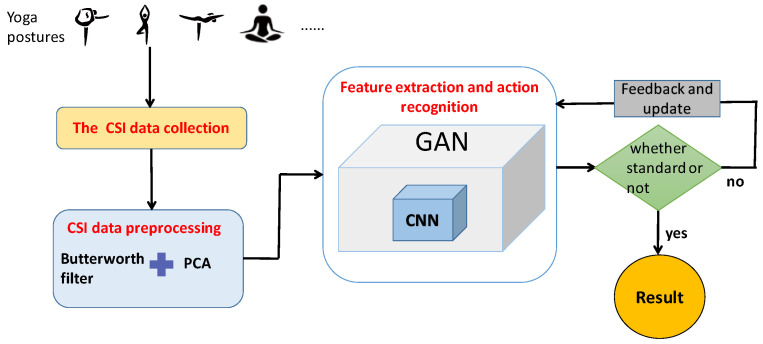
Overview of WiPg.

**Figure 5 sensors-22-00402-f005:**
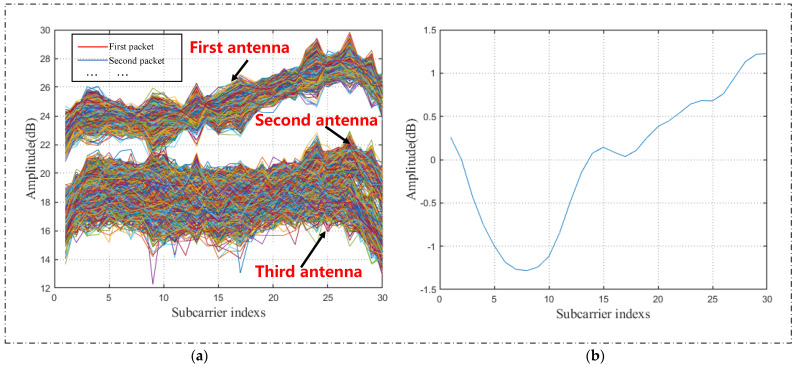
CSI waveform of the yoga pose. (**a**) Original CSI waveform; (**b**) CSI waveform after preprocessing.

**Figure 6 sensors-22-00402-f006:**
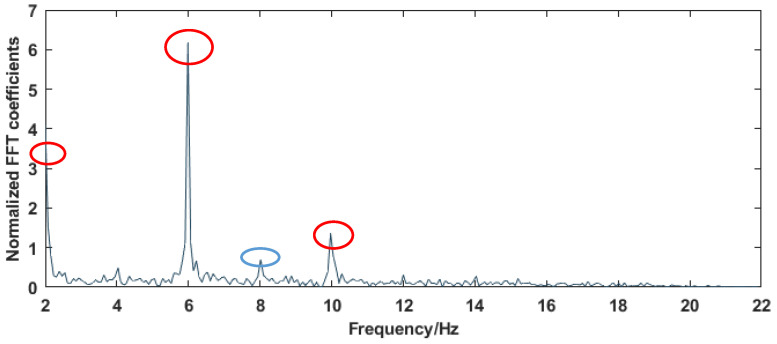
POSE TWO’s FFT curve.

**Figure 7 sensors-22-00402-f007:**
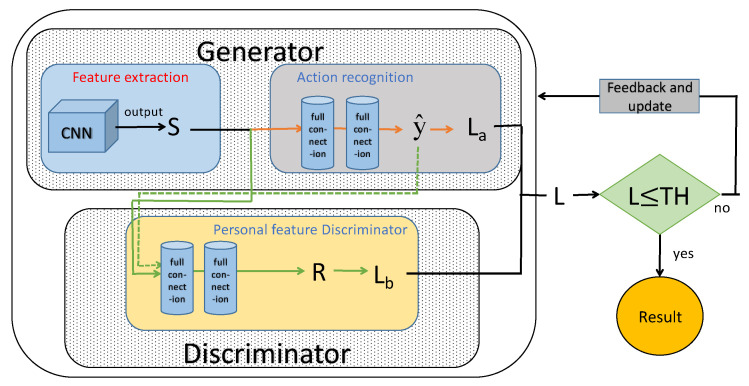
Action-recognition method in WiPg.

**Figure 8 sensors-22-00402-f008:**
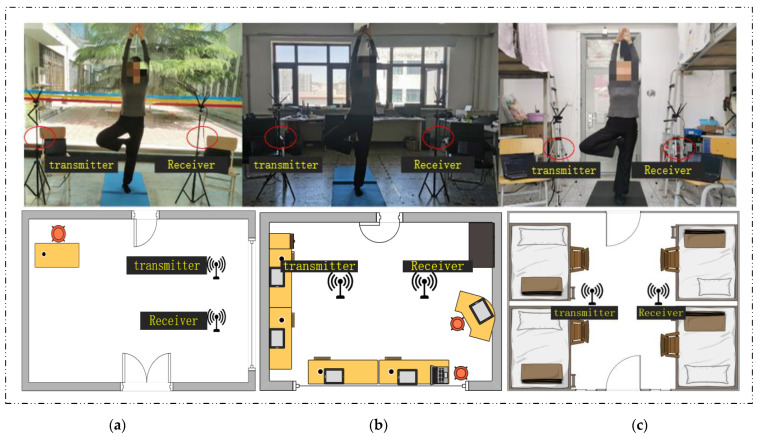
Experimental environment. (**a**) Yoga room; (**b**) laboratory; (**c**) dormitory.

**Figure 9 sensors-22-00402-f009:**
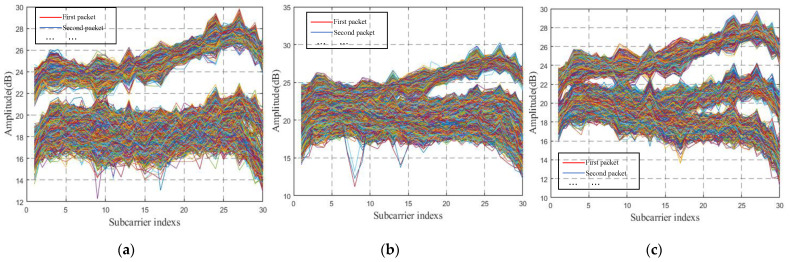
Original CSI signals of different people doing POSE THREE in Figure 1. (**a**) Experimenter A-POSE THREE; (**b**) experimenter B-POSE THREE; (**c**) experimenter C-POSE THREE.

**Figure 10 sensors-22-00402-f010:**
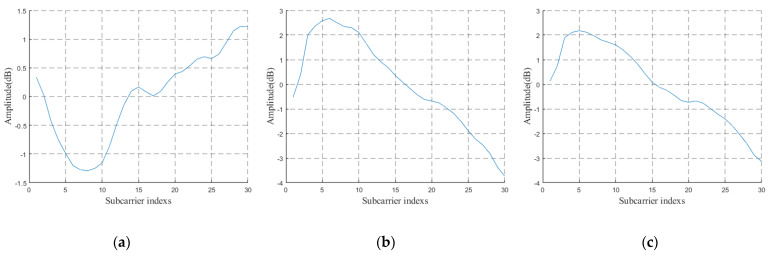
Preprocessed CSI signals of different people doing POSE THREE in Figure 1. (**a**) Experimenter A-POSE THREE; (**b**) experimenter B-POSE THREE; (**c**) experimenter C-POSE THREE.

**Figure 11 sensors-22-00402-f011:**
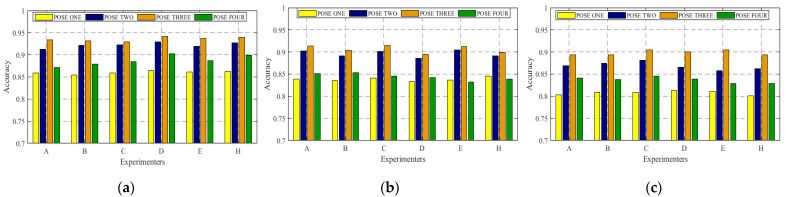
Recognition rate of different people practicing different actions in different experimental scenes. (**a**) Yoga classroom; (**b**) laboratory; (**c**) dormitory.

**Figure 12 sensors-22-00402-f012:**
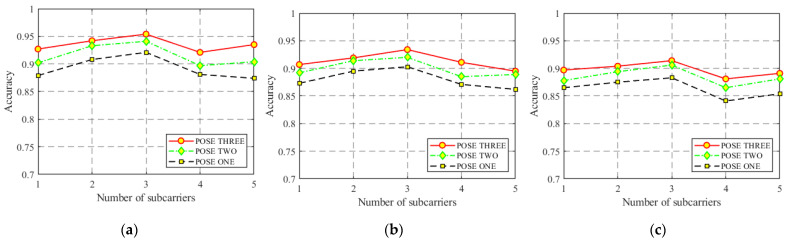
Accuracy of different numbers of subcarriers. (**a**) Yoga classroom; (**b**) laboratory; (**c**) dormitory.

**Figure 13 sensors-22-00402-f013:**
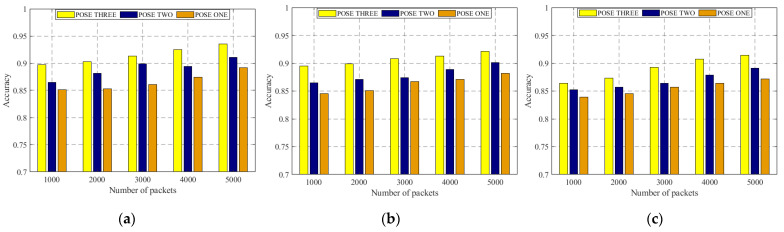
Accuracy of different CSI packet numbers. (**a**) Yoga classroom; (**b**) laboratory; (**c**) dormitory.

**Figure 14 sensors-22-00402-f014:**
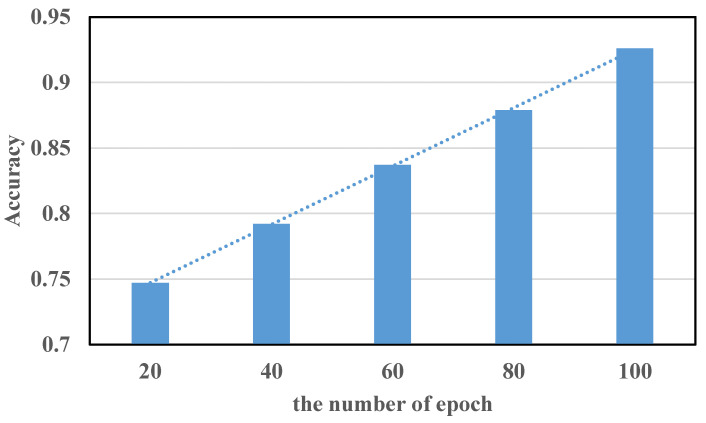
Different numbers of epochs.

**Figure 15 sensors-22-00402-f015:**
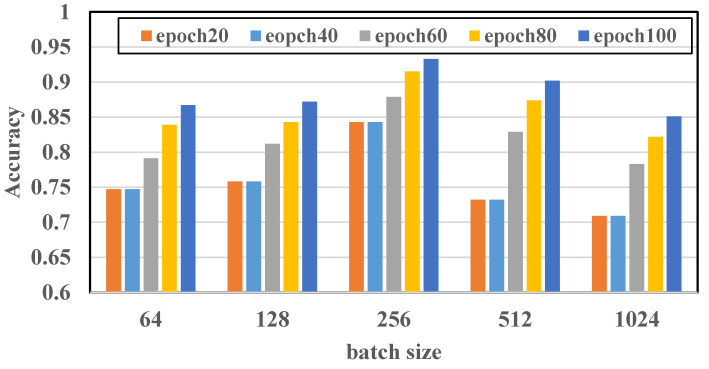
Different batch sizes.

**Figure 16 sensors-22-00402-f016:**
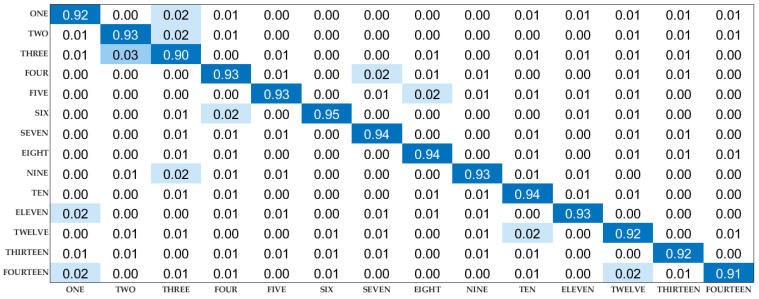
Confusion matrix for synthesizing experimental results.

**Figure 17 sensors-22-00402-f017:**
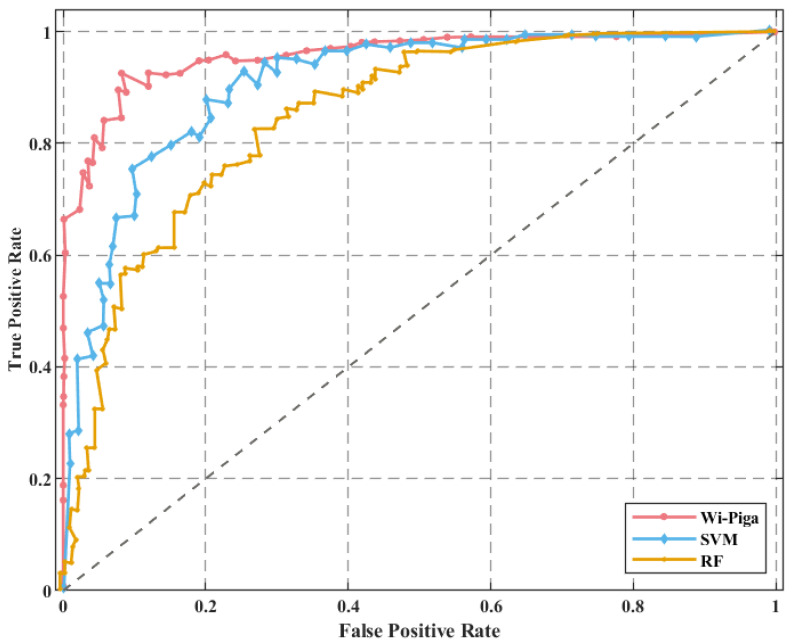
ROC curve comparison in the yoga classroom.

**Figure 18 sensors-22-00402-f018:**
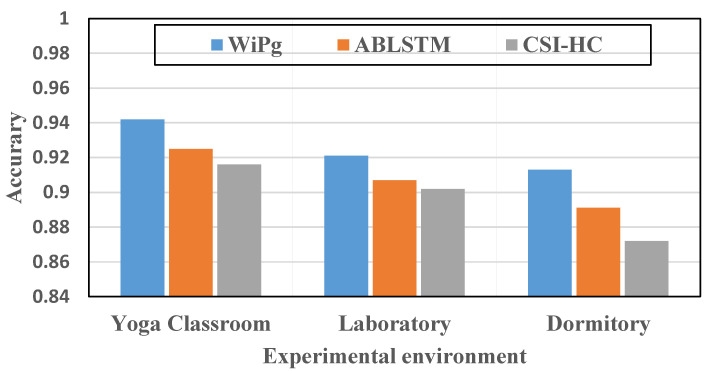
Robustness comparison.

**Figure 19 sensors-22-00402-f019:**
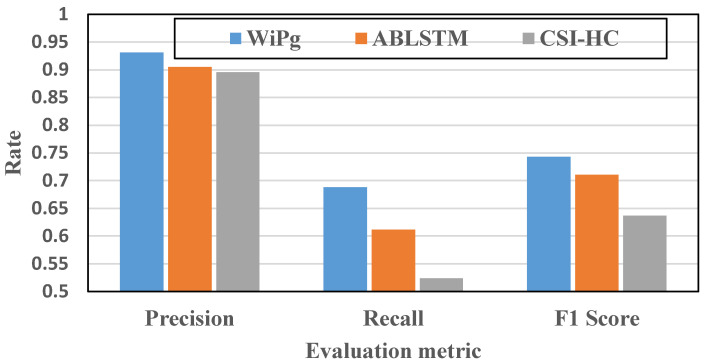
Comprehensive evaluation of classification performance.

**Table 1 sensors-22-00402-t001:** Body characteristics of the experimenters.

Experimenter	Gender	Height (m)	Weight (kg)	BMI
A	Female	1.57	57	23.13
B	Female	1.62	54	20.57
C	Female	1.66	51	18.50
D	Female	1.75	52	16.97
E	Female	1.80	63	19.44
F	Male	1.69	56	20.84
G	Male	1.73	70	23.39
H	Male	1.77	80	25.53
M	Male	1.85	75	21.91
N	Male	1.89	70	19.59

**Table 2 sensors-22-00402-t002:** Comparison of different recognition methods.

Different Experimental Scenarios and Methods		Three Evaluation Indexes		
		Accuracy	F1 Score	AUC Area
**Yoga Classroom**	RF	0.899	0.849	0.897 4
	SVM	0.906	0.857	0.923 4
	the method of WiPg	0.933	0.907	0.942 1
**Lab**	RF	0.858	0.845	0.884 3
	SVM	0.891	0.853	0.904 5
	the method of WiPg	0.914	0.876	0.912 5
**Dormitory**	RF	0.834	0.841	0.857 6
	SVM	0.872	0.851	0.894 7
	the method of WiPg	0.899	0.867	0.909 5

## Data Availability

Data is contained within the article.

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
