# Peer review of "WiPg: Contactless Action Recognition Using Ambient Wi-Fi Signals"

_sensors, 2022, doi:10.3390/s22010402_

Round 1
Reviewer 1 Report
There are some key comments to be addressed. A major revision is recommended.
Comment 1. Abstract:
(a) Elaborate the importance of the research topic.
(b) Highlight the key findings/results of the research topic.
Comment 2. Keywords:
(a) More precise keywords should be included to better reflect the scopes of the paper.
Comment 3. Section 1 Introduction:
(a) Some references [x] are formatted with superscript.
(b) Literature review, include more latest works (2019-Now). Also, summarize the methodology, results, and limitations of the existing works.
(c) Summarize the research contributions of the paper.
Comment 4. Section 2 and 3:
(a) The methodology of the proposed work is not clearly presented. The major focus of current manuscript is background information of the baseline algorithms.
Comment 5. Section 4 Experiment and Analysis:
(a) Justify the setting of the environment and how it matches the reality.
(b) Subsection 4.2, it seems to be the metrics for the performance evaluation, instead of “accuracy standard”.
(c) Figure 8, legends are missing.
(d) Elaborate the statement “However, with the increase of the number of epochs, the update times of the weights in the neural network are also increasing, and the curve changes from underfitting to overfitting.”
(e) Ensure high resolutions for all figures.
(f) Comparison between proposed work and existing works is missing.
Comment 6. Discuss the limitations of the proposed work and suggest future research directions.
Reviewer 2 Report
In this paper, the authors present a method by which to detect human actions using ambient Wi-Fi signals. I thought this paper was very interesting and well written, though perhaps a bit contrived in the experimental setup. More specifically, the major issue I see with the work is that the ambient Wi-Fi emitters were placed very close to the human subject and not overly representative of a realistic setup. However, as a proof of concept on RF fingerprinting in this application space, I think this is ok. In addition, I thought the use of the GAN for removing human size deviations was interesting.
Minor Comments:
- Would be good to see more specific prior art in RF fingerprinting technology. This was a particular application of interest for Ultra-Wide Band technologies in the past.
- "Yoga" spelled incorrectly in Table 2
- Typically shouldn't use the word "Reference" before citation numbers.
- BP acronym used before defined
- GAN is introduced in 2.3 and then later on explained why it is needed in WiPg. Suggest explaining the why before the deep dive to improve readability
- It is a bit confusing what the confusion matrix of Fig. 14 represents. Seemingly, the developed approach is to detect whether a pose is correct or not, not classify the correct pose. Looking at Fig. 3, it isn't clear where the classification comes from before judging if the pose is correct. Ultimately, the author should better describe how Fig. 14 and Fig. 15 are generated from the approach illustrated in Fig. 3. Is the underlying assumption that the approach knows which pose is being attempted?
Reviewer 3 Report
This paper focused on the issue of motion recognition with different body types. A personnel-independent action recognition model called WiPg which is built by Convolutional Neural Network (CNN) and Generative Adversarial Network 13 (GAN) is proposed in this paper. Experiments show that WiPg has good recognition performance and can achieve human independence. This research work is interesting for the human activity recognition research society, and the experimental results are discussed suitably. However, this paper has several limitations and the standard is not enough, and address the following items would result in a good paper,
- The literature review is not thorough about the application and the contributions. To highlight the contributions, it suggests reorganizing the section of the related work. At least, for each contribution, it should be novel and meaningful according to a thorough literature review. In the literature analysis, it is recommended to read the following works and consider to discuss their similar applications in the introduction and discussion, Multi-sensor Guided Hand Gestures Recognition for Teleoperated Robot using Recurrent Neural Network, Towards Teaching by Demonstration for Robot-Assisted Minimally Invasive Surgery.
- It is recommended to present in the first section so that it can highlight the specific scope of this article. The meaning of the assessment experiment should be highlighted.
- Maybe it is better to discuss the possibility to improve the accuracy using deep learning in the introduction, for example, A Smartphone-Based Adaptive Recognition and Real-Time Monitoring System for Human Activities, A Multimodal Wearable System for Continuous and Real-time Breathing Pattern Monitoring During Daily Activity.
- Figure 5 is not clear and we would suggest adding more details and labels to make it clear.
- There should be a further discussion about the limitation of the current works, in particular, what could be the challenge for its related applications.
- To let readers better understand future work, please give specific research directions.
Round 2
Reviewer 1 Report
Authors have significantly improved the quality of the paper. I have some follow-up comments.
Comment 3. Section 1 Introduction:
(a) Some references [x] are formatted with superscript.
Reply : Thank you very much for your valuable suggestions on the revision of our manuscript, which further improves the quality of our article. All the authors of the manuscript would like to express their heartfelt thanks to you! We have formatted the references [x] with superscript according to the revision opinions of the reviewer teacher, and see Section 1 Introduction for details
Follow-up comment: The comment has not been addressed. Please refer to the template.
(b) Literature review, include more latest works (2019-Now). Also, summarize the methodology, results, and limitations of the existing works.
Reply : Thank you very much for your valuable suggestions on the revision of our manuscript, which further improves the quality of our article. All the authors of the manuscript would like to express their heartfelt thanks to you! We have revised and written the Literature review according to the revision opinions of the reviewer teacher, and see the manuscript for details.
Follow-up comment: Please ensure the methodology, results, and limitations of the existing works are summarized.
Comment 5 (d) Figure 8, legends are missing.
Reply : Thank you very much for your valuable suggestions on the revision of our manuscript, which further improves the quality of our article. All the authors of the manuscript would like to express their heartfelt thanks to you! We have revised and written the article according to the questions and opinions of the expert teachers. The details are as follows:
Follow-up comment: In the figures, there are multiple curves with different colours, what do they represent for?
(f) Ensure high resolutions for all figures.
Reply : Thank you very much for your valuable suggestions on the revision of our manuscript, which further improves the quality of our article. All the authors of the manuscript would like to express their heartfelt thanks to you! We have revised and written all Figures according to the revision opinions of the reviewer teacher, and see the manuscript for details.
Follow-up comment: The comment has not been addressed.
Reviewer 3 Report
The authors have addressed all of my concerns.
Hence the current version can be accepted now.
Author Response
Please see the attachment

This manuscript is a resubmission of an earlier submission. The following is a list of the peer review reports and author responses from that submission.